# Exploring Factors and Impact of Blockchain Technology in the Food Supply Chains: An Exploratory Study

**DOI:** 10.3390/foods12102052

**Published:** 2023-05-19

**Authors:** Abubakar Mohammed, Vidyasagar Potdar, Mohammed Quaddus

**Affiliations:** School of Management and Marketing, Curtin University, Perth, WA 6845, Australia

**Keywords:** blockchain, food supply chain, adoption framework, challenges, impacts

## Abstract

Blockchain technology (BCT) has been proven to have the potential to transform food supply chains (FSCs) based on its potential benefits. BCT promises to improve food supply chain processes. Despite its several benefits, little is known about the factors that drive blockchain adoption within the food supply chain and the impact of blockchain technology on the food supply chain, as empirical evidence is scarce. This study, therefore, explores factors, impacts and challenges of blockchain adoption in the FSC. The study adopts an exploratory qualitative interview approach. The data consist of Twenty-one interviews which were analyzed using thematic analysis techniques in NVivo (v12), resulting in identifying nine factors classified under three broad categories (Technology—complexity, compatibility, cost; Organization—organization size, knowledge; Environment—government support, competitive pressure, standardization, and compliance) as the most significant factors driving blockchain adoption in the FSC. In addition, five impacts were identified (visibility, performance, efficiency, trust, and value creation) to blockchain technology adoption. This study also identifies significant challenges of blockchain technology (interoperability, privacy, infrastructure conditions, and lack of knowledge). Based on the findings, the study developed a conceptual framework for blockchain adoption in food supply chains. The study adds to the corpus of knowledge by illuminating the adoption of blockchain technology and its effects on food supply chains and by giving the industry evidence-based guidance for developing its blockchain plans. The study provides full insights and awareness of blockchain adoption challenges among executives, supply chain organizations, and governmental agencies.

## 1. Introduction

The population of the Earth is projected to reach 8.5 billion by 2030, 9.7 billion by 2050, and 10.9 billion by 2100 [1,2]. The likelihood of exceeding the Earth’s capacity to regenerate is considerable due to rapid population expansion and harmful anthropogenic influences in nature; thus, aggressive objectives for reducing harmful environmental impacts have been set worldwide [3,4,5]. In addition, having a significant impact on climate change, the food industry is one of those most negatively impacted by its effects [6], such as through glasshouse gas emissions [7], while also having a significant impact on it [8]. In our society and economy, the food supply chain (FSC) plays a significant role [9,10]. FSC is a complex system comprising several stakeholders, such as consumers, farmers, industries, manufacturers, distributors, retailers, and the government [11,12,13]. All these stakeholders have different roles in the FSC [14,15,16], from farmed crops to consumers, due to information asymmetry [11]. Globalization has increased the complexity of FSCs, in production, shipping, and other operations [17,18]. The complexity of the supply chain, however, raises the risk of product fraud and a failure in confidence among the supply chain participants [19,20]. FSC’s globalization has also contributed to several significant challenges in the overall food systems, including food security and food fraud [21,22]. Food security, as defined by the Food and Agriculture Organization (FAO), is the capacity of every individual to always have access to enough food that is safe, nutritious, and sufficient for them to lead active lives [23]. Food fraud is characterized as a set of intentional or unintentional actions taken for financial gain. Food fraud incidents have emerged as a challenge in the world’s food system, such as the 2008 Chinese milk scam [24] and the massive “food theft” controversy in India [25]. The United Kingdom (UK) also experienced a horsemeat controversy in 2013 [26]. The same year, 15 European nations and Hong Kong were impacted by an egg contamination incident [27], which impacted consumer attitudes and beliefs in the food market [22]. Other findings on food insecurity, made were available for about 150 countries in 2014 and 2015, indicate that sub-Saharan Africa has the greatest rates of food insecurity because of conflicts’ role in causing serious starvation [28,29]. Over half of the adult population in that area suffered moderate to severe levels of food insecurity, and one-fourth of them experienced severe levels. Southern Asia had the second-highest incidence, with 12% of all adults experiencing serious food insecurity [30], due to the lack of trust, transparency, and ineffective food traceability [31]. The food production process must be controlled to ensure food safety, from the raw ingredients to the table. The food sector has since made significant investments in information systems and cutting-edge technologies to improve the management of food products. Researchers and industry experts have been looking towards technology that might help with FSC traceability issues for the past few years. An illustration of a radio frequency identification (RFID) tag aids in tracking a product’s history to ensure the traceability and visibility of a specific product [32]. A “quality-sustainability decision support system (QSDSS)” approach is presented in [33] as a means of assuring food products’ quality and safety. Similarly, [34] created a conceptual framework to address food products’ quality, safety, sustainability, and logistics effectiveness along the supply chain. Despite the improvements in some areas of the food industry, food security remains a significant worldwide issue and has gained increasing attention in recent years. BCT has emerged as an innovative tool that can address significant industry problems [35,36] based on its prospective advantages and benefits [37]. Gartner predicted that the market value generated by BCT will hit USD 176 billion by 2025 and USD 3.1 trillion by 2030 [38]. Similarly, WinterGreen stated it will reach USD 60.7 billion by 2024 [39]. By 2025, the global market for BCT adoption is anticipated to increase from USD 4.6 billion to USD 20.3 billion [40]. International Data Corporation (IDC) estimated that blockchain solutions would grow by 75% in 2022 [41]. A report from multinational corporation PricewaterhouseCoopers (PwC) [42] indicated that “600 executives from 15 territories, 84% say their organizations have at least some involvements with blockchain technology”. The outcome demonstrates that the financial services sector continued to lead in the adoption of BCT. PwC discovered blockchain technology has potential in various industries, including energy, utilities, and healthcare. The country with the most advanced BCT is the United States (US). The Chinese president made a statement in 2019 [43] promoting the adoption of blockchain in China, and as a result, it is anticipated that China will soon take the lead [42]. The blockchain has been greatly impacted by this internationally.

The food industry has acknowledged BCT’s enormous potential, and its application has been emphasized. It promises to enhance conventional food supply chain operations [22]. The BCT’s immutable feature enhances FSC performance and lowers the fraudulent activities in the FSC, which include intangible assets that cannot be measured physically, including contamination and product purity. BCT is a shared public and private database of all digital assets across blockchain partners using distributed ledger technology [44]. Blockchain is already used in the government, mining, healthcare, education, and supply chain [45,46]. Some major corporations, including IBM, Walmart, and Tsinghua University, have investigated BCT to address China’s food safety challenges and enhance food traceability throughout the supply chain [47]. Despite blockchain’s unique and valuable features, it has yet to establish itself in the food sector [22,48,49]. Our study tries to close this gap by examining BCT’s factors, impacts, and challenges in the FSC. In light of the study’s goals, the following research questions are examined:What are the factors influencing the adoption of BCT into the FSC?What are the impacts of BCT on the FSC?What are the significant challenges of BCT adoption in the FSC?

The rest of the paper is organized as follows. Section 2 discusses blockchain in food supply chains. Section 3 provides the theoretical foundation. Section 4 defines the methods. Section 5 presents the findings and discussions of the study, and Section 6 describes the conclusions.

## 2. Blockchain in the FSC

Many studies have demonstrated the potential of blockchain applications in the FSC. According to estimates, the worldwide blockchain market for the food supply chain was worth USD 128.87 million in 2020. By 2025, it is anticipated to grow at a CAGR of 47.1% and reach USD 886.18 million [50]. The global market value of BCT in the food and agriculture sector was expected to increase from USD 32.2 million in 2017 to USD 1.4 billion by 2028. It was also anticipated to increase by 42.85% in Europe between 2018 and 2028, 40.42% in North America, 7.85% in the Asia-Pacific, and 48.33% annually in the rest of the world [51]. Another prediction of using BCT in the FSC and agricultural industries was to increase from USD 41.9 million in 2018 to USD 1.4 billion by 2028 [52]. This shows that BCT can be helpful in FSC and help with various tasks, including supply chain tracking and monitoring [17]. Blockchain has gained attention in scholarly studies. However, the acceptance of BCT in the FSC is scarce [53,54]. Many studies have offered insightful opinions on how blockchain could enhance the FSC. To track food and determine real-time food traceability, ref [55] connected IoT with blockchain utilizing Hazard Analysis and Critical Control Points (HACCP), which keep track of every item in the food chain. 

Furthermore, [56] also used blockchain and the IoT to create an integrated system for tracing eggs from farms to forks, a method for managing the supply chain that is visible and traceable [57] to confirm product quality [58]. Their findings demonstrated that participants could attest to a product’s high quality. Additionally, ref [59] proposed a food safety and traceability framework in the Vietnamese dairy industry. In [60], the authors also identify BCT-based solutions for resolving food traceability problems and emphasize the benefits and drawbacks of putting BCT-based traceability systems into practice in their analysis of BCT’s characteristics and potential applications. Additionally, they suggested an architecture design framework and a flowchart for appropriateness application analysis for BCT-based food traceability systems to aid researchers and practitioners in putting BCT-enabled food traceability systems into practice. In [61], a BCT framework was proposed for halal FSC. They also stressed the significance of supply chain integration and food laws under the halal FSC as critical factors in the success of BCT. In addition, academics continue to push for in-depth studies into the adoption of blockchain in certain industries to address problems in those industries and demonstrate blockchain’s benefit for enhancing supply chain performance [62]. There are presently not many empirical studies in the literature examining the blockchain adoption factors and the impact of blockchain in this domain.

## 3. Theoretical Foundation

This study, therefore, draws together technology, organization, environment (TOE), and the resource-based view as the basis for investigating blockchain adoption in food supply chains. TOE is a theory that examines the factors that drive adoption. This study also explores the impact and challenges of blockchain within food supply chains. This study integrated TOE and resource-based views as underpinning theories to propose a conceptual framework. Past studies have provided frameworks to explain the adoption of blockchain in the supply chain. For instance, ref [63] studied blockchain adoption using the unified theory of acceptance and use of technology (UTAUT) and theory of acceptance. The deployment of blockchain in the US–India supply chain was outlined in their blueprint. This study shows how enabling conditions, social influence, and performance expectations can impact widely adopted blockchain technology. The authors in [64] conducted a survey based on the TOE framework to determine how small-to-medium-sized enterprises (SMEs) in Malaysia could utilize blockchain. These findings show how factors such as cost, relative advantage, complexity, and competitive pressure greatly impact people’s behavior, including their intention to adopt blockchain. While doing so, ref [65] presented a multi-stage model for the diffusion of blockchain technology that took into account a number of theories, including the diffusion of innovations theory (DOI), resource-based view, dynamic capability, technological adoption model, and institutional approach. In [66], the research resulted in the proposal of a research model that integrates three theories: the theory of planned behavior (TPB), the technological readiness index (TRI), and the technology acceptance model (TAM).

Meanwhile, ref [37] developed a framework for supply chain performance dimensions after identifying essential core adoption elements like speed, risk minimization, flexibility, affordability, and sustainability in a case study. As observed in the literature, enough has been made in the previous studies; however, none has combined TOE and the resource-based view to study factors, impacts, and challenges of BCT within the FSC. Therefore, this study proposed TOE alongside the resource-based view to enable further understanding of the adoption of BCT within the FSC.

## 4. Materials and Methods

The research approach should be determined by the research’s objectives, as each methodology has a unique way of gathering and analyzing data [67]. The use of case studies in information systems (IS) research has been very prevalent [68]. Over the past few decades, the interpretive research discipline has made tremendous advancements. The case study is also appropriate when the boundaries between the phenomena under inquiry and the circumstances are unclear. This study employed a qualitative strategy to address the research questions. The qualitative technique used in the study makes this possible by gathering precise information and generating fresh insights into the phenomenon [69,70,71]. There are three reasons why the qualitative approach is used in this study. First, qualitative research is proper when exploring a new domain [72]. Second, qualitative analysis best serves exploratory research in which the crucial variable is still developing [72]. Third, qualitative research, in line with the interpretative stance used in the study’s design and execution, may shed light on and explain the adoption of blockchain in the food industry. Blockchain technology is in its early stages, when insufficient or little research is available. Qualitative studies use a variety of data collection methods. This study used interviews for data collection.

### 4.1. Interviews

Research interviews can be broadly divided into three types based on how the questions are framed: unstructured, semi-structured, and structured. Structured interviews employ a series of predefined queries, provide no room for exploring, and lose depth. Unstructured interviews are those in which there are no pre-planned questions or topics. Semi-structured interviews pursue a medium path, requiring the development of a list of pre-planned questions to be investigated [73]. The primary data were gathered through semi-structured interviews, in which certain questions were prepared beforehand to support and direct the interviewee while keeping the conversation on the topic. Contrary to a positivist interview, no set format was required, allowing the dialogue to progress and new questions to be produced as the interview went on [74]. For identifying participants for our semi-structured interviews, we reviewed public sources such as news articles and industry reports to identify companies in the food industry that have initiated the adoption of blockchain. Popular news items, websites, personal industry connections, and media communications were all used to find relevant participants. In addition to professional social media platforms such as LinkedIn, emails were used to connect with senior managers and secure their consent to participate in the study. All the participants were asked several questions regarding the factors and impact of BCT adoption in the FSC. The participants were chosen based on their specialist knowledge and expertise with BCT and the supply chain. All the interviewees identified to participate in the study have worked on blockchain projects in the food sector and came from different countries, including Australia, New Zealand, India, Canada, and the U.S. The study also used snowball sampling, which involves using existing networks to find competent managers who satisfy the requirements for participation. The COVID-19 pandemic necessitated conducting the interviews online using the Zoom platform. This had benefits in protecting the health of the researcher and participants but drawbacks because it is preferable to conduct in-person interviews to enable observation of facial expressions and body language [75]. Table 1 outlines the interview participants’ demographics, and Table 2 shows the participants’ profiles. Twenty-one (21) interviews were conducted, each lasting between 30 and 50 min (see Table 2). No translation was required for the interviews, and all were in English.

### 4.2. Data Analysis

Since qualitative research frequently generates massive amounts of data, its analysis is often more difficult and time-consuming than quantitative data [74]. Thematic analysis, a typical coding technique for qualitative research, was used to conduct the qualitative data analysis for this study [76]. Handling and comprehension of the vast quantity of data using this technique were achieved by recognizing, deciphering, and presenting themes derived from the empirical data. The interviews lasted for a total of 739 min, and 130 pages of transcription were produced. The researcher produced the transcript. All transcripts were coded and reviewed to ensure the data references were accurate. NVivo 12 was used to code the text files from each transcribed interview and to support the researcher in ensuring that coding was not carried out automatically without analytical thought. The data were analyzed using a six-step theme analysis procedure outlined in [76], a systematic inductive method, as shown in Table 3. These coding strategies let the researcher stay receptive to the participants’ stories. These themes were subsequently identified, given names, and separated into ten sub-themes, which assisted the author in providing broad concepts and creating a structure.

### 4.3. Quality of Data Collected

The option of retaining anonymity must be provided for everyone participating in a research study [74]. Individual participants in the qualitative element of this study chose to remain anonymous but agreed to let the researcher know the study’s sponsor. The importance of organizations that decided to participate became increasingly evident when they could be identified. However, as it is a researcher’s responsibility to ensure that respondents cannot be recognized, personal information such as names cannot be disclosed [74]. Revealing the interviewee’s precise position might also be problematic because it could allow for identifying personal data.

Construct validity, internal and external validity, and reliability were evaluated to guarantee the high quality of the study. By combining evidence from several sources, following evidence chains, and member checking, construct validity can be attained. Pattern matching and other well-established analytical methods can establish internal validity, whereas analytical generalization is used to prove external validity. Table 4 describes the criteria used to achieve reliability and validity in this study.

## 5. Findings and Discussion

This section presents the findings by answering the three research questions about factors, impacts, and challenges affecting BCT adoption in FSC. These factors, impact, and challenges were merged to develop a conceptual framework for BCT adoption in the food supply chain, as presented in Figure 1.

### 5.1. Blockchain Technology Adoption Factors

This section answers the first research question on factors affecting blockchain technology adoption. The study extracted the themes and sub-themes from the thematic analysis of the transcribed responses. The factors of adopting BCT were then organized using thematic analysis. The question posed to respondents was, “What factors encouraged the adoption of blockchain technology?”. Responses were analyzed using the TOE framework, and themes were discussed based on technological, organizational, and environmental aspects. We discuss each element separately.

#### 5.1.1. Technological Factors

The study explores participants’ views on the technological factors involved in adopting BCT in the FSC [77]. This study found that complexity, compatibility, and cost can influence BCT’s adoption in FSCs. The following subsection explains the identified factors based on the respondents’ responses; Table 5 illustrates the example of codes.

Complexity is *“the degree to which an innovation is seen as relatively hard to understand and apply”* [78]. The degree of complexity reflects how challenging the innovation or new technology is to comprehend, implement, and use [78]. An established factor for technology adoption is the perceived complexity of the innovation. An organization’s possibility of accepting innovation decreases as its complexity increases. This factor is also one of the three elements of the DOI theory (i.e., relative advantage, compatibility, and complexity), which are the most relevant factors for examining technology adoption [77]. Previous studies found that complexity [79,80,81] negatively affects technology adoption [82]. The responders emphasized the need for enterprises need to exercise caution while implementing blockchain technology. There is a lot of friction still in blockchain as a technology regarding users being able to access it directly. The responses evidence this; P16 stated, *“What, I think, you know, our role is to provide an excellent application to our supplies and to make users interact with that application rather than interacting directly with the blockchain. So, it is just not user-friendly technology for our broad group of the supply chain”*. The complexity of the ecosystem and the range of stakeholders means that collaboration is necessary to produce value from the technology; P11 mentioned, *“Not all supply chain equipment is suitable for or ready for rapid digitalization. And even digitalized technologies, in terms of the information they collect, may not be in a cloud-ready state that would enable ready access from a blockchain point of view”.* P5 stated further, *“I think spreading literacy about technology will help food supply chain partners to come on board, try, and implement new solutions, that will reduce their perception on the complexity of blockchain”.* This view was supported by one participant, P9, who said, *“We discovered that scaling up the solution in a sustainable fashion was one of the main hurdles with blockchain technology”.* Complexity may result from several factors, including the fact that blockchain is a problematic technology. P2 mentioned, *“How to get beyond the complexity of the food ecosystem is the first obstacle facing food supply chain, who are just entering the market”.* The complexity of blockchain, as well as the potential difficulties associated with it, might lower the adoption to the targeted food sectors.

Compatibility *is “the extent to which an innovation is viewed as consistent with the existing values, past experiences, and needs of potential users”* [78]. The perception of BCT is how blockchain can be compatible with an organization’s needs, goals, infrastructure, and processes [83]. Compatibility is an essential factor for the acceptance of innovation [84]. High compatibility has been viewed as an enabler for the uptake of innovations [84]. According to adoption diffusion research, where potential adopters view suitable innovation as “less unclear,” it is more likely to be adopted quickly. The authors in [85] demonstrated similarly how a compatible innovation increases integration within a company, such as with supply chain partners. P17 said, *“To consider the business cases and determine what the beneficial business case may or may not be to understand an existing technological implementation”*. How can BCT be compatible with the company’s existing technology? This is a question of whether blockchain solutions can connect with the company’s existing food supply chain systems. P12 said, *“So that is an important question because many of these companies might be using some software, maybe an ERP or some other software, which they might have already spent a lot of money to get into work”*. So, the first question is, will blockchain replace that software, or is blockchain credible enough to replace that? P12 added, *“So, my answer to that is blockchain is not replacing any software. Blockchain adds credibility to the information generated by this software”*. P4 voiced the same opinion on blockchain compatibility, stating, *“I think our system will be capable enough to pull and push the data from that existing engine and provide an update to the farmers”*. This entails interacting with the system and making that data transfer happen.

Cost is a significant factor in adopting innovation [77]. Cost can be categorized into two aspects: direct and indirect cost. The direct cost is related to obtaining the technology, while indirect costs are created by maintenance, implementation, and use. The cost is essential for technology adoption, so it is hardly considered a roadblock. The cost has always had an impact on adoption. As also stated in [77], this cost must be regarded as in an organization’s decision before adopting the technology [86]. Blockchain is relatively affordable from a commercial and development perspective. Blockchain, however, comes with additional charges. There are processing expenses, for instance, if you wish to run an application against the blockchain. One interviewee mentioned that blockchain development platforms had hidden costs; P9 stated, *“We must be very mindful of the cost structure and the dynamics of physical environments and the industry to decide on what the most appropriate technology interventions need to be”*. According to another participant, blockchain has no cost-effectiveness issues, as highlighted by these comments; P15 mentioned, *“I know that certain blockchain solutions have become cripplingly expensive. We provide super low-cost digital infrastructures for food suppliers, and data is accessible. Accessing our API is open like we are trying to be as open source as possible. The only thing where we encounter a challenge around costs is making payments. And that is because now, we must move on to traditional payment rails. So, we encounter the same costs you would have with any other payment provider, but that becomes like an additional layer when delivering finance and making payments. So, it is still reasonably low cost”*. P3 found, *“The blockchain solution’s price must be competitive to attract users like farmers. The company has seen significant cost savings because of this”*. Blockchain allows businesses to comply with financial regulations in real-time.

#### 5.1.2. Organizational Factors

The organization’s qualities are described by organizational factors. The elements that have been thoroughly investigated for technology adoption are discussed below. BCT requires an organization to have a solid resource foundation because it is a complicated and expensive technological breakthrough. Resources are better able to manage risks and payoffs as a firm grows. Therefore, compared to smaller firms, large organizations are more ready to adopt developing technology to gain an advantage over their rivals. The size of the business strongly influences the adoption of blockchain-based technology. As a result, it is defined as a determinant in the study model to examine the impact of organizational size on the corporate adoption of blockchain supply chains. Table 6 illustrates the example of codes.

Organization size: The size of the organization is an influential factor [87]. For example, large organizations are more willing to accept new technologies than small organizations due to their flexibility and ability to soak up the risk [88,89]. P18 stated, *“Obviously, the bigger the organization, the more potential it must justify a significant capital cost implementation”*. In many supply chains, large incumbents resist applying new technologies such as blockchain until they are ready to roll over into a blockchain environment. P14 gave an example: *“If you are an industry leader, you can exploit market dominance without spending large amounts of capital to sustain that position”*. In this study, size is preferable to organization size. The number of cattle, the number of people working on the farm, or the farm’s income can all be used to measure the size of a farm. Most studies concur that the most prevalent indicator of organizational size is the number of employees. 

Knowledge: There is, generally, limited awareness of blockchains, and often the knowledge that people would have picked up in exposure to the discussions on the technology is misleading. P13 described, *“Certainly, as a new technology, there is just knowledge, a learning curve. So, you know, not everybody involved in blockchain has the level of knowledge, especially since the end users receiving the value may not have the most knowledge of blockchain”*. Another participant, P7, said, *“The adoption is relatively low. We have been around for about two you plus years. So, people are becoming more familiar with it. They did not associate when we first came to market. Everyone was thinking of Bitcoin, or some cryptocurrency, is that dark side of the net. And so now we rarely get that type of question. Now we get different questions about how this technology enables sharing kind of what a permission blockchain means, so the questions have evolved so that people are becoming more familiar with Blockchain technology. There’s, quite frankly, more information published about the technology. So, people are reading more about it and becoming more educated”*. Furthermore, P7 added, *“And then the other aspect is many pilots are still running in the supply chain. However, it’s still early for them to say they’re in full rollout mode”*. Enterprises see BCT as an integrated technology, which means they see it as a complement to already-in-place technical solutions.

#### 5.1.3. Environmental Factors

The environment is the physical and social aspects that directly influence how people behave while making decisions in organizations [90]. Environmental factors can be classified as either internal or external environments. External environmental factors are those aspects of the environment outside the control of the organization’s management that might endanger or benefit the organization [91,92]. The external environment consists of those “global” external elements beyond an organization’s control yet which are crucial to its operation and decision-making processes. In contrast, the internal environmental factors are organizational traits. As a result, this study explicitly considers the external environment rather than the “environment” [93]. Table 7 illustrates the example of codes.

Government Support can significantly impact the adoption of blockchain technology. Governments may also offer financial incentives and pilot programs to encourage technical innovation. The government can play a significant role in the adoption and diffusion of innovations through information provision, research and development policies and facilities, incentives, building and enhancing the infrastructure, running pilot projects, offering tax breaks, and providing consulting and counselling services [94,95,96]. There are some external requirements that, through acting as dual-edged factors, have been discovered to influence food industries’ decisions to implement BCT. It would be riskier for individual farmers to adopt new technologies and practices without government involvement. P11 stated, *“To my point before, there is some way where I think the government, you know, has... a role to play. And you know, it should not just be you, and I know the big guys at the banks have done some blockchain stuff to support some food industries”.* P3 added, “*I think there is an opportunity for the government to set up a blockchain task force. Scale the marketplace. Many people like me are doing projects with SMEs [small to medium enterprises]*”. Blockchain is a technology that requires competitors to compete with; P4 describes, “*It is not just that we can take advantage of great technologies like blockchain, which have a lot of potential benefits, but also that we can agree on standards to have better governance across supply chains*”. This participant added that there is a lot of movement, and governments are interested in this. P19 highlighted, *“I still think there is a lot of scepticism. So, I think the awareness is certainly growing. But I think there is a lot of scepticism in the agriculture industry. In part, that is because, for decades and decades, people have gone to the agriculture industry with new technologies and said, ’implement this technology’”.* P14 recalled, *“I think we will see more and more interest over the next six months from state governments interested in expanding exports into Asian food sectors”.* Looking at traceability technologies, some more of these will come out soon enough. However, because governments are interested, the respondent stated, *“I think that is also spurring even further interest in the agriculture industry. Things like the national blockchain roadmap, which came out earlier this year, are a big signpost for many industries that this is happening. But, of course, there are a lot of obstacles and challenges in adopting this technology” (P8)*. Adopting the technology will not happen out of thin air but because of a rule or regulation. Competitive pressure is described as *“the degree of pressure faced by the companies from competitors inside the industry”* [97]. Organizations are encouraged to research to thrive and remain competitive in the market. Competitive pressure is a key factor influencing the adoption of new technologies. Blockchain-based solutions offer more efficiency and transparency, which give the food industry essential competitive benefits [65]. P9 shared, *“So because blockchain is getting gaining popularity so, possibilities are that your competitor is contemplating using Blockchain, and you do not want to stay behind them. So, in that sense, you also want to go ahead and stay with them to stay in tune with what the market is doing”*. From the perspective of participant P2, *“There is a bit of peer pressure. This technology is fascinating, and everybody wants to implement that. Some people and food industry can appreciate the value, and some companies do not see any immediate value. Still, everyone knows this technology has potential, and they are starting to get open to it”*. Standardization: In addition to controlling blockchain usage, it is essential to standardize the terminology. Standardization is important to increase the advantages of great technologies like blockchain, which have many potential benefits, and that agreement on standards ensures better governance across food supply chains. P17 emphasized, *“Blockchain has no regulation as such, and the regulation will come into the picture when there are smart contracts, and maybe there will be questions on the validity of smart contracts as legal documents. But most of the use cases have not reached that maturity level. And so, it remains to be seen how that will shape up, but sooner or later, maybe the industry will have some framework standard that will become the standard for the industry”*. The respondent further stated that this was necessary to facilitate more collaborative effort across the supply chain more generally. P13 said, *“So, it’s not just so that we can take advantage of great technologies like blockchain, which have a lot of potential benefits, but also so that we can agree on standards so that we can have better governance across supply chains”.* Compliance: Another expectation is compliance in complex supply chains, particularly in the FSC. There is a need for extensive compliance for all sorts of things, ranging from animal safety, food safety, and human safety to environmental impacts. One participant, P7, commented, “*Create new regulatory regimes that can be inadequate cost and data-driven, almost to a point where compliance becomes an automatic feature of ongoing operations*”. The ability to achieve compliance and report on compliance cost-effectively is beneficial to those who must comply with something to be able to operate lawfully.

#### 5.2. Impact of Blockchain Technology

What are the impacts of BCT on the FSC?

Visibility: It is possible to define supply chain visibility as having *“access to high-quality information that explains diverse demand and supply elements”*. This concept is frequently enhanced by the capacity to recognize and validate crucial data (such as identification, location, and status) of a product as it moves through the supply chain. Others refer to the ability to determine a product’s path as traceability. Despite some ambiguity in the terminology, researchers concur that supply chain visibility is linked to several advantageous operational and financial outcomes, such as decreased uncertainty and disruption risks, lower inventories, and improved responsiveness (*“In general, supply chain management and logistics have several main objectives, but achieving supply chain visibility is one of them. It is, moreover, a significant outcome of essential supply chain procedures like external integration and knowledge sharing,*” said P12). The information would be more trustworthy, and there would be no information asymmetries, ultimately boosting food visibility. The participants argued that blockchain technology’s primary reason is its visibility. As stated by P11, *“Businesses can deliver information to clients when they are visible, which gives them a chance to do so”*. Performance: Blockchain improves supply chain quality management by lowering costs and sharing information with the right partner, impacting industry performance and sustainability. With the approval of others in the supply chain network, information transparency enables individual organizations to track the flow of products. This improves industry performance by reducing the likelihood of corrupt practices and fake goods, as P20 described: *“I think that supply chain governance performance could be enhanced by blockchain technology”*. Managers and policymakers must take the initiative to establish a platform for fostering collaboration to improve performance. P18 supported this by saying, *“The requirement for improving the fresh food supply chain’s performance is coupled with the technological aspects of the blockchain in several ways”*. Efficiency: BCT has a significant and advantageous role in enhancing businesses’ operational effectiveness since it solves various issues with data sharing and resource integration in multi-party collaboration. *“The capability to do this in the short term is to be able to enhance operational efficiency within a lot of organisations. But operational efficiency needs to be related to network efficiency,”* said P12. Blockchain is not necessarily the best technology to use if it is going to be internalized and solve internal problems. P12 further added, *“But if we’re looking at solving network problems across the whole of supply chain problems, then blockchain could be a great technology. But to solve those problems, it’s not good enough to have great technology; you must have collaborative effort and the supply chain participants’ leadership to want to be able to share enough information to benefit from implementing that technology”*. Trust: Transparency and traceability are correlated with firms’ readiness to establish customer trust in their communications and foster an awareness of their business operations. P5 stated, *“When one considers the numerous potentials of a blockchain, it is about acquiring better control and the capacity to enable the appropriate data”*. Value creation relates to business case propositions and different uses of information within complex ecosystems. Supply chains have other demands for information. P9 stated, *“What I would call productivity benefits. And they largely go towards the ability of somebody engaged in a value creation process in the supply chain to achieve that value creation with fewer inputs”.* Many agricultural producers are interested in improving the quality of their products because it will change the value and price of that product. According to P14, “*Value growth beneficiaries are related to the ability, through data, for the new value to be created and for more earnings from a supply chain activity*”. Furthermore, the results demonstrate that *“Added value through greater transparency and traceability is correlated with firms’ readiness to establish customer trust in their communications and foster an awareness of their business operations”,* as stated by P8.

### 5.3. Challenges of Blockchain Technology

The biggest obstacle to using blockchain is still comprehending what it can and cannot achieve. Its continued evolution as a technology presents another challenge. This indicates that it is still under development and not a complete solution. It should be more evident that blockchain should be used with other developing technologies as a solution. Even though the technology has already been around for ten years, there are still some adoption challenges in the industrial setting. Blockchain adoption is being hindered significantly by a lack of education and awareness. *What are the significant challenges of BCT adoption in the FSC?* Interoperability: The demand for standards and protocols is heightened by the necessity of interoperability throughout the food supply chain. Blockchain-based solutions face difficulties in being adopted because there is no consensus protocol. P16 stated, *“I think blockchain technology is still very young. And it is not a great surprise. I think it is held up to be something that was going to be this. This is a super ground-breaking solution”*. The food industry has been exposed because of the lack of visibility into the supply chain (*“So, one of the problems is the supply chain’s lack of visibility and the incompatibility of visibility”,* said P18). Privacy: Blockchain provides peer-to-peer data transfer via a decentralized network without needing any third party. This study found that privacy is considered another challenge in the food supply chain process. One participant noted this: *“So, for example, privacy solutions are not solved adequately in most blockchains or distributed ledgers”* P15). P9 further stated, *“It is challenging to work with blockchain, and it is not like many solutions are solved with blockchain in the supply chain”*. Infrastructure conditions: P10 described, *“So, if you do not have good internet connectivity to areas, or the cost of communications of data is expensive. You must modify your approach to how you collect that information and how you transmit that data*”. Another participant, P17, said, *“So even though the idea of having ear tags for cattle that can capture all the information you could imagine for cattle anywhere in Australia, theoretically, is fantastic, transmitting that data continually poses challenges. In addition, the ability to maintain energy [for] devices to send information poses data in remote locations”*. Lack of knowledge: Even though BCT is unfamiliar to many people, some are knowledgeable and have recognized its significance. According to the respondents, the biggest obstacle to the widespread usage of blockchain is the general public’s lack of understanding of it. “*People are still learning about it. They do not understand the difference between a blockchain ledger system in a database or how the technology can improve or provide value to the whole ecosystem*”. Although internal BCT knowledge is required, it may not be the only important component influencing the adoption rate. The results demonstrate the importance of comprehending the benefits of adopting BCT. Businesses should view BCT as a solution that adds value to their operations and yields results.

## 6. Conclusions

This study summarizes the present knowledge on blockchain’s factors, impacts, and challenges in the FSC. BCT has been proven to have the potential to transform the FSC based on its potential benefits. Blockchain can improve product traceability and speed up determining the origin of products. Blockchain also provides end-to-end product traceability, tracking food products at every stage of the food supply chain. This study used a qualitative interpretative research approach to collect interview data. After collecting and analyzing the interview data, this study then proposed a conceptual framework for blockchain adoption in the FSC. This framework integrates factors, impacts, and challenges to BCT adoption in the FSC. The study’s findings identified complexity, compatibility, cost (Technology), organization size, knowledge (Organization), government support, competitive pressure, standardization, and compliance (Environment) as the most significant factors driving blockchain adoption in the FSC. This study also discussed the impact of BCT and how blockchain could enhance visibility, performance, efficiency, trust, and value creation in FSC processes, based on the evidence from the interviews. Moreover, it also identified interoperability, privacy, infrastructure conditions, and lack of knowledge as the significant challenges of blockchain adoption.

The study was constrained by the early stages of BCT adoption, making it more challenging to connect with food industries with sufficient background knowledge and experience with the phenomenon. As a result, the organizations taking part in this study maintain varying degrees of BCT adoption experience and expertise, which forces the researchers to engage with and thoroughly study the empirical data. The intention to concentrate on a particular supply chain was constrained by the absence of organizations that had expertise and experience in BCT adoption as well as producing the same product, which is related to the immature stage of BCT adoption in the food industry. As a result, the study concentrates on the food industry. The dearth of academic studies on BCT adoption in general and those relevant to the food industry might be considered a limitation. First, in this study, we only covered the qualitative approach. The literature lacked empirical studies, so a qualitative approach was ideal for our research. Second, we only focused on one general case study, the food supply chain. We did not focus on a specific food supply chain, such as the milk or grain supply chain. However, our research sets the foundation for others to undertake future research that could be more focused on specific product supply chains or investigating a particular factor or factors. Third, all the interviews were conducted online due to COVID restrictions. This might have impacted the quality of interviews, as when interviews are face-to-face, people can be more engaged and provide additional details. However, we ensured that the interviews were as engaging as possible to gather sufficient depth. Fourth, a lot of companies have not implemented blockchain yet, so many of their responses were based on their assumptions. In other cases, those companies that had implemented blockchain only had the platform running for a short period as a pilot. Hence, even they were exploring the benefits and challenges of their implementation. However, the information they provided during the interviews was still very useful in understanding the benefits they perceived of using blockchain, the challenges they faced, and the impacts they observed.

There are many future research directions that we identified based on our research. We list five future research directions that we consider promising for researchers to take this research forward. First, one enquiry for future research is the BCT adoption of a particular product supply chain in the food business. This recommendation is supported by research showing that product attributes and other supply chain circumstances influence firms’ adoption decisions. Therefore, such a study would offer even more detailed perceptions of the adoption process. Second, the additional study could strengthen, investigate, and test the BCT framework described in this study, for instance, undertaking a multiple-case study that enables a more in-depth examination of the justifications for various firms’ adoption choices and strategies. Third, researchers could undertake quantitative studies to validate the research findings presented here. There are still limited empirical studies in the literature, and it would be timely to undertake quantitative studies in the next two to three years. Fourth, in the food supply chain, current studies are still focused on a conceptual level, and some studies have developed and tested a pilot. Future studies can focus on a longitudinal study where the blockchain implementation and use can be researched over two or three years. This will provide additional deeper insights into the challenges and benefits of using blockchain technology. Researchers can also investigate the real-time application of blockchain and assess it from the technological, environmental, and organizational perspectives to understand the challenges and potential impacts (both positive and negative). Fifth, another important research direction is to evaluate the factors, challenges, and impacts in the light of technology fusion, i.e., combining blockchain with other emerging technologies like Artificial Intelligence, Machine Learning, Big Data, Internet of Things, Cloud Computing, and Robotic Processes Automation. Blockchain, on its own, can have technological limitations. For example, one area where blockchain can be challenging is when the data are entered manually. There is a potential that such data can be erroneous or could lead to manual data entry errors. If blockchain is combined with IoT, where IoT sensors are used to capture data and directly store them on the blockchain, the trust in the overall blockchain solution will be much higher. In such cases, researchers can explore new models and frameworks to efficiently integrate different technologies to obtain better results. Sixth, researchers can also investigate the complete life cycle assessment of a particular crop or food product to understand the impact of carbon emission during the growth cycle. Blockchain can be used to record carbon emission-related information throughout the supply chain from farm to fork. Several challenges would need to be resolved during this process, and hence this will be a promising future research direction. Blockchain technology has shown promising potential to improve transparency and traceability in the food supply chain substantially. Building a transparent food supply chain improves the trust among consumers who are calling for better transparency on how and where their food is grown. Consumers are getting more sensitive about what they eat and what they feed to their families. Hence, any technology that can help to improve this situation would be very useful. Blockchain has shown promise to be that technology to meet consumer expectations. In the future, as technology matures, these benefits will be realized, and consumers will be happy and confident. The study adds to the corpus of knowledge by illuminating the adoption of blockchain technology and its effects on food supply chains and by giving the industry evidence-based guidance for developing its blockchain plans. The report provides full insights and awareness of blockchain adoption challenges among executives, supply chain organizations, and governmental agencies.

## Figures and Tables

**Figure 1 foods-12-02052-f001:**
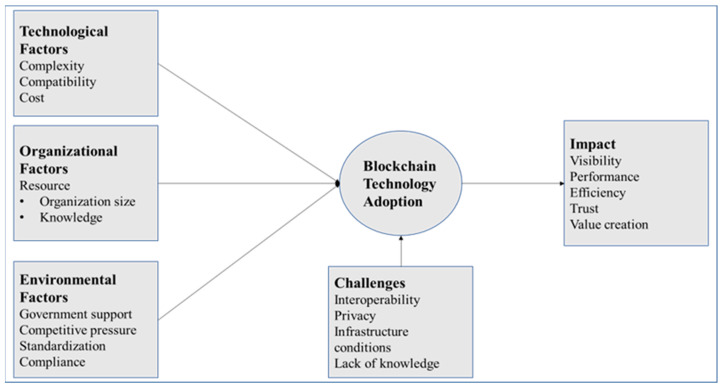
Conceptual framework of blockchain technology adoption in the food supply chain.

**Table 1 foods-12-02052-t001:** Participant’s demographics.

Position	Senior Management	12
	Project Manager	5
	Developer	4
Gender	Male	15
	Female	6
Interview Time	Minimum Minutes	21
	Maximum Minutes	48
	Mean Minutes	36.24
Nationality	Australia	11
	USA	4
	Canada	1
	India	2
	New Zealand	3

**Table 2 foods-12-02052-t002:** Participants’ profile.

Code	Position	Gender	Experience	Length
P1	Project Manager	Male	10+ years	31 min
P2	Solution Architect	Male	5+ years	40 min
P3	Chairman	Male	10+ years	48 min
P4	Management Scientist	Male	5+ years	42 min
P5	CEO and Founder	Male	15+ years	27 min
P6	Software Engineer	Male	4+ years	27 min
P7	Product Management Advisor	Female	5+ years	44 min
P8	Solution Architect	Male	5+ years	37 min
P9	CEO and Founder	Female	15+ years	21 min
P10	CEO and Founder	Female	10+ years	48 min
P11	CEO and Founder	Male	10+ years	36 min
P12	CEO and Founder	Female	10+ years	44 min
P13	Founder	Male	15+ years	36 min
P14	CEO	Male	20+ years	32 min
P15	Technical Analyst	Male	15+ years	30 min
P16	Supply Chain Manager	Male	10+ years	23 min
P17	CEO	Male	20+ years	41 min
P18	Solution Architect	Male	5+ years	32 min
P19	CEO	Female	5+ years	37 min
P20	Director	Male	10+ years	29 min
P21	Project Supervisor	Female	3+ years	34 min
Total				739 min

**Table 3 foods-12-02052-t003:** Data analysis process.

Phase	Description
Step 1: Familiarization of data	Transcription, repeated reading, and taking down initial thoughts
Step 2: Generating initial codes	Systematic coding of essential elements throughout each data collection
Step 3: Search for themes	Putting together all the data relevant to each potential theme and
Step 4: Reviewing themes	figuring out if the themes have anything to do with the coded extracts Examining how well the themes relate to the complete set of data
Step 5: Defining and Naming themes	Ongoing review to enhance the narrative analysis as a whole and the specifics of each subject
Step 6: Report	The final examination of chosen extracts and the selection of vivid, engaging extract samples. Lastly, creating a scholarly summary of the analysis by connecting the analysis to the research question and the literature

**Table 4 foods-12-02052-t004:** Data trustworthiness.

Criteria	Description of Strategy
Transferability	Details on the participants, the method, and the setting of the current study
Credibility	Method Triangulation: Multiple data collecting triangulation, interviews.
Dependability and Confirmability	Ensuring consistency in the questions requested of the participants and using qualitative data analysis software NVivo

**Table 5 foods-12-02052-t005:** Example of coding for technological factors.

Illustrative Quotes	Examples
Complexity	*What, I think, you know, our role is to provide an excellent application to our supplies and to make users interact with that application rather than interacting directly with the blockchain. So, it is just not user-friendly technology for our broad group of the supply chain (P16)*
	*How to get beyond the complexity of the food ecosystem is the first obstacle facing the food supply chain, which is just entering the market. (P2)*
Compatibility	*So that is an important question because many of these companies might be using some software, maybe an ERP or some other software, which they might have already spent a lot of money to get into work (P12)*
	*I think our system will be capable enough to pull and push the data from that existing engine, provide an update to the farmers (P4)*
Cost	*We must be very, very mindful of the cost structure and the dynamics of physical environments and the industry to decide on what the most appropriate technology interventions need to be (P9)*

**Table 6 foods-12-02052-t006:** Example of coding for organizational factors.

Illustrative Quotes	Examples
Organization size	*If you are an industry leader, you can exploit market dominance without spending large amounts of capital to sustain that position (P14)*
Knowledge	*Certainly, as a new technology, there is just knowledge, a learning curve. So, you know, not everybody involved in blockchain has the level of knowledge, especially since the end users receiving the value may not have the most knowledge of blockchain (P13)*
	*And then the other aspect is many pilots are still running in the supply chain. However, I would say it’s still pretty early for them to say they’re in full rollout mode (P7)*

**Table 7 foods-12-02052-t007:** Example of coding for environmental factors.

Illustrative Quotes	Examples
Government Support	*To my point before, there is some way where I think the government, you know, has... a role to play. And you know, it should not just be you, and I know the big guys at the banks have done some blockchain stuff to support some food industries (P11)*
	*I think there is an opportunity for the government to set up a blockchain task force.* *Scale the marketplace (P3)*
	*It is not just that we can take advantage of great technologies like blockchain, which have a lot of potential benefits, but also that we can agree on standards to have better governance across supply chains (P4)*
Competitive pressure	*There is a bit of peer pressure. This technology is fascinating, and everybody wants to implement that. Some people and the food industry can appreciate the value, and some companies do not see any immediate value (P2)*

## Data Availability

Data is contained within the article.

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
