# Peer review of "Exploring Factors and Impact of Blockchain Technology in the Food Supply Chains: An Exploratory Study"

_foods, 2023, doi:10.3390/foods12102052_

Round 1

Reviewer 1 Report

There is an issue with the coherence of the article's ideas in line 31-32. The sentences seem disconnected from the preceding and following sentences. For instance, the mention of increasing population in line 32-33 doesn't seem related to the discussion on food fraud and food security. It is suggested that additional articles could help provide more context or support for the ideas being presented. Some relevant articles that may aid in strengthening the article's arguments have been provided: https://doi.org/10.1016/j.jclepro.2023.136894;  

The acronym TOE needs to be expanded before its first usage to ensure that readers understand its meaning. The in-text citation format in section 3.4 needs to follow the MDPI referencing style. Thirdly, the arrows in Figure 1 are too faint and should be changed to black to improve visibility.

The presentation of sections 4.1.1 - 4.1.3 needs to be improved, possibly through the use of a tabular form, to make it easier for readers to understand the information presented.

While the article is focused on the application of BCT in the food sector, it is not clear if all 21 participants in the study are from the food sector. Additionally, it seems that the interviews conducted and the answers given by interviewees do not reflect a food sector focus.

Overall, there are major flaws in the article that need to be addressed before it can be considered for publication. The article requires improvement, such as clarity, formatting, and consistency. Addressing these issues can help to strengthen the article and make it more suitable for publication.

The article consists of sentences that is difficult to understand. For instance,  "Due to fewer issues in the food industry, food security has grown to be a significant worldwide issue and has gotten greater attention recently." This sentence may be confusing due to the use of negative phrasing ("fewer issues") and the unclear relationship between the first and second clauses. To improve clarity, the authors may want to consider revising the sentence to better convey their intended meaning. For example, they could rephrase it as follows: "Despite improvements in some areas of the food industry, food security remains a significant worldwide issue and has garnered increasing attention in recent years." Overall, the authors need to go through the whole manuscript to eliminate such sentences and to know the importance of clear and concise writing to ensure that readers can easily understand the ideas being conveyed.

Reviewer 2 Report

Manuscript ID: foods-2350649

Title: Exploring Factors and Impact of Blockchain Technology in the Food Supply Chains: An Exploratory Study

This topic seems interesting. The authors propose a sentiment classification method of social network text based on AT-BiLSTM model in a big data environment. The proposed method is compared with three other methods using the same dataset by simulation experiments. The results show that the author's method obtains the optimal result. However, there are some issues to be resolved at this stage based on the following comments:

  1. The abstract part is written too briefly and must be revised to highlight the innovation of the article. The abstract must be a concise yet comprehensive reflection of what is in your paper. Please modify the abstract according to “motivation, description, results and conclusion” sections.
  2. The paper cites many works; it is clear that an important amount of work has been studied. However, the paper lacks a thorough comparison of the cited works and a detailed discussion of the open issues related to this field.
  3. Ablation experiments should be added in the experimental part to prove the effectiveness of the module designed by the author.
  4. Talk more about the efficiency of the network?
  5. Is the work that the authors compared in the experimental section typical of work in the field? Do the algorithms for the author's comparative experiments have public codes?
  6. Some key model parameters are not mentioned. The rationale for the choice of the set of parameters should be explained in more detail. Have the authors experimented with other sets of values? What are the sensitivities of these parameters on the results?
  7. The conclusion must be revised to highlight the innovation of the article. At the same time, some specific data can be added to show the advantages of the algorithm.
  8. The format of the references requires careful proofreading. The page numbers of many references indicate inconsistencies. Moreover, I suggest the authors read and consider the studies performed by scholars such as Tirkolaee et al., Weber et al., and their groups to add more readability to the intro, LR and outlook sections.
  9. Please polish the English. I found several errors.

The authors must polish the English.

Round 2

Reviewer 1 Report

The manuscript has gone through lots of revision. However, there are few things which need to be addressed such as the sections 'Limitations' and 'Future work' need to be part of Conclusion. Also, the referencing should be as per MDPI style.
